# M1 Macrophage-Biomimetic Targeted Nanoparticles Containing Oxygen Self-Supplied Enzyme for Enhancing the Chemotherapy

**DOI:** 10.3390/pharmaceutics15092243

**Published:** 2023-08-30

**Authors:** Jiayi Zhang, Bing Gu, Shimiao Wu, Lin Liu, Ying Gao, Yucen Yao, Degong Yang, Juan Du, Chunrong Yang

**Affiliations:** 1Department of Pharmacognosy, College of Pharmacy, Jiamusi University, Jiamusi 154007, China; zhangjiayi1998@163.com (J.Z.); 15765361644@163.com (B.G.); miaoastq@163.com (S.W.); yaoyucen68@163.com (Y.Y.); 2Department of Pharmacy, Shantou University Medical College, No. 22 Xinling Road, Shantou 515041, China; simonell817@163.com (L.L.); gaoying19930527@126.com (Y.G.)

**Keywords:** tumor microenvironment, M1 macrophage, tumor hypoxia, catalase, cancer treatment

## Abstract

Tumor hypoxia is considered one of the key causes of the ineffectiveness of various strategies for cancer treatment, and the non-specific effects of chemotherapy drugs on tumor treatment often lead to systemic toxicity. Thus, we designed M1 macrophage-biomimetic-targeted nanoparticles (DOX/CAT@PLGA-M1) which contain oxygen self-supplied enzyme (catalase, CAT) and chemo-therapeutic drug (doxorubicin, DOX). The particle size of DOX/CAT@PLGA-M1 was 202.32 ± 2.27 nm (*PDI* < 0.3). DOX/CAT@PLGA-M1 exhibited a characteristic core-shell bilayer membrane structure. The CAT activity of DOX/CAT@PLGA-M1 was 1000 (U/mL), which indicated that the formation of NPs did not significantly affect its enzymatic activity. And in vitro drug release showed that the cumulative release rate of DOX/CAT@PLGA-M1 was enhanced from 26.93% to 50.10% in the release medium of hydrogen peroxide, which was attributed to the reaction of CAT in the NPs. DOX/CAT@PLGA-M1 displayed a significantly higher uptake in 4T1 cells, because VCAM-1 in tumor cells interacted with specific integrin (α4 and β1), and thereby achieved tumor sites. And the tumor volume of the DOX/CAT@PLGA-M1 group was significantly reduced (0.22 cm^3^), which further proved the active targeting effect of the M1 macrophage membrane. Above all, a novel multifunctional nano-therapy was developed which improved tumor hypoxia and obtained tumor targeting activity.

## 1. Introduction

The tumor microenvironment (TME) featured with hypoxia, acidosis, and dense extracellular matrix is closely related to the resistance of tumors to various therapies. In particular, tumor hypoxia has been regarded as one of the critical causes of poor effects for various cancer treatment strategies [1]. To date, several strategies have been developed to alleviate tumor hypoxia, including enhancing intratumor blood flow, delivering oxygen into tumors, oxygen production in situ, and targeting hypoxia-induced factors [2,3]. Elevated concentrations of intracellular hydrogen peroxide (H_2_O_2_) are a biochemical characteristic of tumor cells. The high concentration of H_2_O_2_ in TME offers new opportunities to design smart self-responsive platforms for alleviating hypoxia-associated resistance to tumor therapy [4]. O_2_ generation using tumor-abundant, endogenous H_2_O_2_, such as self-supply, has been considered an attractive strategy to alleviate tumor hypoxia and improve therapeutic efficacy [5].

Catalase (CAT), a specific catalytic enzyme that is used to decompose H_2_O_2_ into O_2_, has been explored for the fabrication of nano-theragnostic for tumor treatment by the in-situ production of O_2_ combined with other therapeutic approaches [6]. CAT exhibits excellent intrinsic biocompatibility and biodegradability, and an extremely high enzyme catalytic ability [7]. Unfortunately, as an exogenous enzyme, free CAT tends to be rapidly degraded by protease and is prone to being denatured in complex physiological environments, resulting in a significant limitation of its function as an O_2_ self-supplying agent [8]. Han et al. combined porous platinum nanospheres with GOX to achieve synergistic cancer treatment. Porous platinum nanospheres can decompose exogenous and endogenous H_2_O_2_ to produce O_2_ and contribute to GOX-catalyzed glucose oxidation [9]. Peng and his colleagues manufactured CAT-based liposome encapsulated photosensitizers and doxorubicin for chemical photodynamic therapy. H_2_O_2_ in tumors can be decomposed by CAT to produce O_2_, thereby improving the efficacy of chemical PDT [10]. However, most O_2_ self-production systems still face limitations such as low catalytic efficiency, a lack of selectivity for cancer cells, or time-consuming synthesis. Therefore, it is of great significance to develop a simple treatment method that can effectively self-generate O_2_ with a good tumor cell targeting ability and a low systemic toxicity.

Macrophages that are prevalent in the TME have good tumor-homing ability, which is directly related to tumor progression and metastasis [11]. Macrophage membranes contain α4 integrin and CC chemokine receptor 2 (CCR2), making them ideal candidates for tumor targeting, and have been successfully used in bionic systems for drug delivery to tumor or inflammatory sites [12,13]. However, living cells are not suitable as drug carriers due to their disadvantages such as short survival time and difficult storage. Using live cell membranes as carriers not only preserves the functional proteins on the membrane surface but also has allows for a controllable size and high stability. Various NPs, encapsulated in macrophage membranes, showed excellent stability and tumor-targeting ability [14]. Activated M1 macrophages can kill tumor cells with tumor targeting and phagocytosis capabilities [15]. Therefore, in contrast to previous studies by researchers which used non-polarized macrophage membranes, we use M1 macrophage membrane-encapsulated NPs with enhanced tumor targeting to deliver drugs for cancer therapy.

Here, we aim to design a specifically targeted, decomposable/biodegradable, and deeply tumor-penetrating nanoplatform for anti-cancer combination therapy (Figure 1). The macrophage membrane was harvested from RAW 264.7 cells, as they highly express integrins α4 and β1, which have a strong binding affinity for VCAM-1 of mouse mammary carcinoma (4T1) [16]. On the other hand, by taking advantage of excessive amounts of H_2_O_2_ within the TME, the CAT could induce the decomposition of tumor endogenous H_2_O_2_ and generate O_2_ in situ, alleviating tumor hypoxia. In this work, we attempted to develop M1 macrophage membrane-coated biomimetic NPs for combined relieved TME and chemotherapy for anti-tumor treatment. PLGA NPs were first prepared for co-loading doxorubicin (DOX) and CAT, and then encapsulated with M1 macrophage membranes to obtain the NPs of DOX/CAT@PLGA-M1. This drug delivery system could achieve the active targeted enrichment of tumor sites in vivo. Further, the well-protected catalase within the NPs acted as an oxygen generator to decompose endogenous H_2_O_2_ and produce O_2_, thus improving tumor hypoxia and TME, and combining with chemotherapy to improve anti-tumor therapeutic effects. The proposed NPs, encapsulating M1 macrophage membranes, can effectively target tumor sites, combining improved TME with chemotherapy and the advantage of eliminating tumor hypoxia.

## 2. Materials and Methods

### 2.1. Materials, Cell Lines, and Animals

#### 2.1.1. Materials

Doxorubicin Hydrochloride and polyvinyl alcohol (PVA) were supplied by Aladdin (Shanghai, China). PLGA was purchased from Daigang Biomaterial Co., Ltd. (Jinan, China). CAT was purchased from Yuanye Biotechnology (Shanghai, China). The catalase assay kit was obtained from Nanjing Jiancheng Bioengineering Institute (Nanjing, China). Fetal bovine serum (FBS) was obtained from Solarbio (Beijing, China). CCK-8 assay kit and Hoechst 33,258 were purchased by Beyotime (Shanghai, China), and 1,10-Dioctadecyl-3,3,30,30-tetramethylindotricarbocyanine iodide (DiR) was obtained by Meilun Biotechnology (Dalian, China). All other reagents were analytical reagent grade.

#### 2.1.2. Cell Lines and Animals

RAW264.7 cells were obtained from the bone marrow of Balb/c mice, and maintained in DMEM medium containing 10% FBS and 1% streptomycin/penicillin at 37 °C in humidified 5% CO_2_ incubator. Murine breast cancer cell line cells (4T1 cells) were cultured in RPMI 1640 medium containing 1% streptomycin/penicillin and 10% FBS in a 37 °C humidified 5% CO_2_ incubator.

Female Balb/c mice (6~8 weeks) were obtained from Beijing HFK Bioscience Co., Ltd. (Beijing, China), and were housed with ad libitum food/water under specific pathogen-free conditions. All animal experiments were approved by the Experimental Animal Ethics Committee of Shantou University.

### 2.2. Preparation and Characterization of DOX/CAT@PLGA and DOX/CAT@PLGA-M1

#### 2.2.1. Preparation of DOX/CAT@PLGA

The DOX/CAT@PLGA nanoparticles (DOX/CAT@PLGA) were prepared by method shown in [17]. Briefly, PLGA (20 mg) and DOX (2 mg) were dissolved in 2 mL methylene chloride, and emulsified by sonication (200 W, 10 min) at 4 °C. The CAT (8 mg) was added into PVA solution (1%, *w/v*). Subsequently, the primary emulsion was added to the PVA solution and sonicated for 3 min to form a double emulsion. The organic solvent was removed by rotary evaporation. DOX/CAT@PLGA was obtained by centrifuging at 12,000× *g* rpm for 15 min and washing 3 times with deionized water. According to the same protocol, DOX@PLGA was prepared as control without loading CAT.

#### 2.2.2. M1 Macrophage Membrane Extraction and Characterization

The macrophage (RAW264.7) cells were induced to become M1 macrophages by adding lipopolysaccharide at a final concentration of 1 μg/mL for 24 h. The cells were collected and added to TM buffer (pH = 7.4, 0.01 M Tris buffer containing 0.001 M MgCl_2_), and mixed with 1 M sucrose to a final concentration of 0.25 M sucrose. Then, the cells were broken up by a sonicator and centrifuged at 4 °C (2000× *g*, 10 min) to obtain the supernatant. After the supernatant was centrifuged again at 4 °C (3000× *g*, 30 min), the cell membrane pellet was obtained [18].

A western blotting study was performed to investigate proteins present on the M1 macrophage membrane. The indicated cells were washed with cold PBS and then lysed in the cell lysate (20 mM Tris-HCl, pH 7.4, 150 mM NaCl, 1% Triton X-100) with protease and phosphatase inhibitors for 30 min. Protein concentrations were determined using a BCA protein assay kit (Pierce). An equal amount of protein from each sample was loaded to 8% SDS-PAGE (sodium dodecyl sulfate—polyacrylamide gel electrophoresis) and transferred to PVDF membranes (Millipore Sigma, Merck KGaA, Darmstadt, Germany), then incubated with primary and secondary antibodies as indicated. Immunoreactive bands were visualized using an ECL system (Amersham Biosciences, PA, USA) [19].

#### 2.2.3. Preparation of DOX/CAT@PLGA-M1

The extracted M1 macrophage membranes were used to coat the DOX/CAT@PLGA by fusing macrophage membranes with the DOX/CAT@PLGA and extruding repeatedly via an Avanti mini extruder [20]. In brief, the M1 macrophage membranes were mixed with novel prepared DOX/CAT@PLGA solution, sonicated at 4 °C for 5 min, and slowly and continuously extruded several times with a micro-extruder containing a polycarbonate porous membrane to obtain DOX/CAT@ PLGA-M1.

#### 2.2.4. Characterization of DOX/CAT@PLGA and DOX/CAT@PLGA-M1

The size and morphology of DOX/CAT@PLGA and DOX/CAT@PLGA-M1 NPs were determined using a transmission electron microscope (TEM, H-7650, Hitachi, Japan). Furthermore, the zeta potential and size distribution were measured at room temperature using a Nano-ZS (Malvern, UK). Lastly, the lyophilized DOX, blank NP, and DOX/CAT@PLGA-M were analyzed by the Shimadzu DSC 60 (Kyoto, Japan) to verify the successful encapsulation of the drug.

#### 2.2.5. Stability of NPs and In Vitro Drug Release

The in vitro drug release profiles were performed by the RC-6 dissolution instrument (China) [21]. Totals of 1 mg/mL of DOX, DOX/CAT@PLGA, and DOX/CAT@PLGA-M1 were added in dialysis (3500 Da), dispersed to PBS and PBS containing 100 μM H_2_O_2_ at 37 °C, and continuously stirred at 100 rpm. At 1, 2, 4, 6, 8, 12, 24, and 48 h, 2 mL of the free drug was measured by UV-VIS spectrophotometer instrument (UV 759S, Shanghai, China) at 480 nm with the blank release medium at the same temperature added. And the accumulated release degree of DOX was calculated. The particle size and zeta potential of the NPs were monitored at different predetermined time points to assess the stability.

#### 2.2.6. Hemolytic Assay

The whole blood samples were centrifuged at 1500 rpm for 6 min to separate red blood cells (RBCs) from serum. RBCs were washed 3 times with PBS and centrifuged at 1500 rpm for 6 min. The RBC suspension and different concentrations of NPs were placed in EP tubes containing heparin and then incubated at 37 °C for 40 min. The supernatant in the EP tube was transferred to a 96-well plate and the absorbance value was measured at 545 nm by a microplate reader (Tecan, Switzerland) [22].

#### 2.2.7. Catalase Activity Assay

A simple method for determination of serum catalase activity and revision of reference range can be found in [23]. H_2_O_2_ (50 mM) was incubated with free CAT and DOX/CAT@PLGA-M1 (0.5 μM) for 1 h at 37 °C, respectively. The ammonium molybdate (32.4 mM) was added and the temperature was lowered to 25 °C to terminate the catalytic reaction of CAT. Finally, the absorbance at 405 nm was determined by a UV-VIS spectrophotometer, to quantify the remaining H_2_O_2_ and calculate the enzyme activity value (U/mL) based on the detection results.

### 2.3. In Vitro Cell Cytotoxicity

Cell viability was measured by a CCK-8 assay kit to assess the cytotoxicity of different NPs. Briefly, 4T1 cells were seeded in 96-well plates (1 × 10^5^ cells/well) overnight at 37 °C. The cells were next disposed with free DOX, DOX@PLGA, DOX/CAT@PLGA, and DOX/CAT@PLGA-M1 at different DOX concentrations (0.1, 0.5, 1, 2, 4, 8 μg/mL), respectively. After 24 h of incubation, the CCK-8 solution (10%) was added and incubated for 4 h. Cell viability was measured by the absorbance detected at 450 nm using a microplate reader.

### 2.4. In Vitro Cellular Uptake

The 4T1 cells (1 × 10^5^ cells/mL) were seeded in 12-well plates incubated for 24 h at 37 °C. The medium was removed and replaced with a fresh medium containing free DOX, DOX@PLGA, DOX/CAT@PLGA, and DOX/CAT@PLGA-M1 for different periods of 1 h and 4 h. After that, the cells were washed with PBS three times and labeled with Honest 33,258 to dye the nuclei. The cells were observed by confocal laser scanning microscopy (CLSM). Then, the cells were collected and measured for quantitative analysis using a flow cytometer (C6 BD Laboratories, Franklin Lakes, NJ, USA).

### 2.5. Wound Healing Assay

Wounding-healing assays were performed to investigate the inhibitory effect of the NPs on the migration capacity of 4T1 cells [19]. In detail, 4T1 cells (2 × 10^5^ cells/mL) were inoculated in 6-well plates and further cultured overnight. Afterward, the cells were scratch-wounded by parallel lines, added to different NPs, and incubated in DMEM medium without FBS for 48 h. At predesigned time points, cell images were observed by an Axio-Vert-A1 (Oberkochen, German) microscope.

### 2.6. In Vivo Assessments of Biodistribution

To investigate the targeting of NPs, DiR-labeled free DiR, DiR/CAT@PLGA, and DOX/CAT@PLGA-M1 were intravenously injected into 4T1 tumor-bearing mice, respectively. After administration for 1, 4, 12, and 24 h, the mice were anesthetized, and the in vivo fluorescence imaging of mice was performed using a FX Pro imaging system (Bruker, Inc., Madison, WI, USA). After 24 h of in vivo imaging, the major organs (mouse liver, spleen, heart, lung, kidney) and tumor tissues were removed, and we performed in vitro fluorescence imaging.

### 2.7. Antitumor Effects and Biosafety of NPs In Vivo

The tumor-bearing mice were randomized to receive by tail vein administration every 2 days (*n* = 6): (1) Saline (Control); (2) DOX; (3) DOX@PLGA; (4) DOX/CAT@PLGA; (5) DOX/CAT@PLGA-M1. The weight and tumor volumes were monitored every 3 days. Tumor volume was measured using the formula: V = biggest diameter × smallest diameter2/2. At the end of the point 21st day, mice were sacrificed, and the lung, heart, spleen, liver, kidney, and tumor tissues for all groups were carefully isolated. The number of macroscopic metastatic nodules on the lung surface was recorded. Histological analysis of the major organs was evaluated by hematoxylin and eosin (H and E) staining to evaluate in vivo toxicity associated with DOX and detect metastasis in the lungs.

### 2.8. Statistical Analysis

Statistical analysis was performed with GraphPad Prism 8. Data are presented as the mean ± SD. For the comparison of multiple groups, the significance was determined using a one-way ANOVA. Statistical significance was set at * *p* < 0.05, ** *p* < 0.01, and *** *p* < 0.001.

## 3. Results and Discussion

### 3.1. Characterization of DOX/CAT@PLGA and DOX/CAT@PLGA-M1

To obtain M1 macrophages, the RAW 264.7 cells were stimulated with LPS for 48 h. The characteristic protein expression of the M1 macrophages (β1 and α4) was measured to evaluate the success of the RAW 264.7 macrophage induction into M1 macrophages [24]. The results of the western blot analysis (Figure 2) implied that, compared to M0 macrophages, the expression of β1 and α4 in M1 macrophages increased significantly, demonstrating the successful induction of the M1 macrophages.

The biomimetic DOX/CAT@PLGA-M1 was prepared by preparing nanoparticles loaded with DOX and CAT by PLGA, and then coating the M1 macrophage membrane with them on the surface. As shown in Figure 3A,B, the particle sizes of DOX/CAT@PLGA and DOX/CAT@PLGA-M1 were (171.35 ± 1.80 nm) and (202.32 ± 2.27 nm), respectively. The particle size distribution, having a narrow polydispersity index (*PDI* < 0.3), indicated good dispersion and uniform particle size. Compared with DOX/CAT@PLGA, the particle size of DOX/CAT@PLGA-M1 increased by 30 nm, indicating that the cell membrane was successfully modified on the surface of DOX/CAT@PLGA. The morphology of the NPs was investigated by TEM, and the images showed that DOX/CAT@PLGA and DOX/CAT@PLGA-M1 were smooth and uniform with a sphere-like shape (Figure 3C,D). DOX/CAT@PLGA-M1 exhibited a characteristic core-shell bilayer membrane structure with a thickness of about 30~50 nm. As illustrated in Figure 3E,F, the surfaces of DOX/CAT@PLGA and DOX/CAT@PLGA-M1 were both negatively charged (−6.73 ± 0.79 mV and −20.03 ± 1.11 mV). The stability of DOX/CAT@PLGA-M1 was investigated by determining the particle size and zeta potential. The results showed that the particle size and potential exhibited no significant change within 10 days, indicating that the NPs possessed good placement stability (Appendix A).

In addition, a DSC analysis was conducted to investigate the physical state of DOX on the NPs (DOX/CAT@PLGA-M1), and is displayed in Figure 4A. For free DOX and the physical mixture of DOX+ CAT@PLGA-M1, we detected the first melting peak of DOX at 209.81 °C. However, no melting peak was observed in the curve of DOX/CAT@PLGA-M1 at the same temperature, demonstrating that the DOX loaded in the NPs was in an amorphous or solid solution state [25].

To evaluate the hemocompatibility of the nanoparticles, we incubated DOX/CAT@ PLGA-M1 with RBCs at 37 °C for 1 h. As shown in Figure 4B, the hemolysis rates of 0.5, 1, and 2 mg/mL DOX/CAT@PLGA-M1 on the RBCs were 0.94%, 1.59%, and 2.38%, respectively, and all were lower than the standard value of 5%. The results indicated that the NPs had good biocompatibility and stability, and were thus suited for intravenous administration. Next, the enzyme activity of CAT was measured to investigate the effect of loading on the nanoparticles. As shown in Figure 4C, compared with free CAT, the enzymatic activity of DOX/CAT@PLGA-M1 was 1000 (U/mL), and the relative activity was 65.99%, indicating that the formation of NPs could not significantly affect the enzymatic activity.

The in vitro drug release assays were measured by the dialysis method (Figure 4D). When the dialysis lasted for 10 h, almost all of the free drugs (97.7%) were released from the dialysis bag, while the cumulative release rate of the drugs loaded in NPs was lower than 40%. The drug release curve of the NPs showed burst release in the first 6 h, followed by sustained and slow release. Of note, in the release medium of hydrogen peroxide, the drug release rate significantly increased. The cumulative release rate of DOX/CAT@PLGA-M1 enhanced from 26.93% to 50.10% within 10 h. This result may be attributed to the reaction and decomposition of CAT in the NPs with H_2_O_2_ to O_2_, thereby promoting drug release.

### 3.2. In Vitro Cellular Level Study

The cytotoxicity and cellular uptake of different NPs were investigated. As shown in Figure 5A, the cytotoxicity of the blank nanoparticles was detected by their cell viability. The results showed that the cell viability of blank NPs was higher than 80% and possessed almost no cytotoxicity. The cytotoxic effects of free DOX, DOX/PLGA, DOX/CAT@PLGA, and DOX/CAT@PLGA-M1 were subsequently detected to evaluate their inhibitory effects on tumor cells (Figure 5B). DOX in NPs significantly enhanced the 4T1 cytotoxicity by altering the cellular internalization. Compared with DOX/CAT@PLGA, DOX/CAT@PLGA-M1 displayed a higher cytotoxicity and reduced cell viability by approximately 1.51 times. This is attributed to the fact that the M1 macrophage membrane of the NPs could actively target the tumor cells, thereby improving the internalization of tumor cells, and resulting in greater cytotoxicity and apoptosis [26].

To demonstrate the effect of M1 macrophage membranes in 4T1’s internalization of DOX NPs, the in vitro uptake of free DOX, DOX/PLGA, DOX/CAT@PLGA, and DOX/CAT@PLGA-M1 in 4T1 cells was investigated (Figure 5C,D). The results of CLSM and flow cytometry showed similar time-dependent trends in the cellular uptake of diverse NPs. The cellular uptake rate of the DOX group was significantly higher than that of the DOX/PLGA group. This was possibly due to the encapsulation effect of the NPs decreasing the drug release, thereby reducing the effective internalization of DOX. DOX/CAT@PLGA accelerated the release of DOX due to the release of O_2_ from CAT, making the cellular uptake slightly lower than that of the free DOX, similar to the drug release in vitro. In particular, the encapsulation of M1 macrophages caused DOX/CAT@PLGA-M1 to display a significantly higher uptake in the 4T1 cells. Therefore, VCAM-1 in tumor cells could interact with specific integrins (α4 and β1) to promote the DOX/CAT@PLGA-M1 uptake by 4T1 cells and specifically target tumor sites [12].

Furthermore, cell scratch tests were performed to investigate the inhibitory effects of different NPs on cell migration (Figure 6). In the measurement of scratch wound healing, the untreated group almost filled the wound after 48 h, while the DOX/CAT@PLGA-M1 group showed significant inhibition of migration to the wound. The free DOX group, the DOX@PLGA group, and the DOX/CAT@PLGA group also showed decreases in migration, with lesser effects than that of the DOX/CAT@PLGA-M1 group at the same dose. The NPs encapsulated with M1 macrophages were visualized to perform the desirable ability of anti-cell migration, which could be attributed to the ability of the NPs to be internalized more effectively by the migrated cells.

### 3.3. In Vivo Biodistribution

The in vivo biodistribution of different NPs in 4T1 tumor-bearing mice was investigated using DiR as a fluorescent probe. The mice were injected intravenously with free DiR, DiR/CAT@PLGA, and DiR/CAT@PLGA-M1, and were monitored by the FX Pro imaging system (Figure 7A). Fluorescence images demonstrated that, 1 h after administration, there was an obvious fluorescence signal of DiR-labeled NPs in the tumor tissues, lasting until 48 h. The efficient deposition of the NPs at tumor sites was evident due to enhanced permeability and retention effects. Notably, DiR/CAT@PLGA-M1 presented much stronger fluorescence signals than those of DiR/CAT@PLGA at the tumor site, due to the active targeting of M1 macrophage membranes. To better demonstrate the improved tumor targeting of M1 macrophage membranes, tumor-bearing mice were sacrificed 48 h after injection, and major organs and tumors were harvested for ex vivo fluorescence imaging (Figure 7B). The intertumoral DiR accumulation rate of DiR/CAT@PLGA-M1 was higher than that of DiR/CAT@PLGA. These results supported that DOX/CAT@PLGA-M1 could transport DOX and CAT to the tumor site and proved the feasibility of the strategy of using biomimetic NPs.

### 3.4. In Vivo Antitumor Efficacy

As illustrated in Figure 8A, tumor-bearing mice were randomly assigned to several groups for treatment and monitoring. The intravenous administration of diverse NPs was performed every three days. The growth of 4T1 tumors was significantly inhibited, as evidenced by representative images of in vitro tumor tissues from the different treatment groups (Figure 8B,C). Meanwhile, the tumor volume change in mice for all groups was monitored and recorded to draw tumor volume change curves. As shown in Figure 8D, moderate anti-tumor activity was observed in the DOX and DOX/PLGA groups, indicating that free DOX is difficult to accumulate in tumor cells for tumor killing. However, DOX/CAT@PLGA exhibited better anti-tumor effects due to CAT producing O_2_ at the tumor site, which improved the combined effect of TME and DOX. Interestingly, compared to the DOX/CAT@PLGA group, the tumor volume of mice in the DOX/CAT@PLGA-M1 group was significantly reduced, with an average tumor volume of only 0.22 cm^3^ after treatment, demonstrating a significant anti-tumor effect, which strongly highlights the active targeting effect of the M1 macrophage membrane during drug delivery. In addition, no significant weight loss was found in the 4T1 tumor-bearing mice during antitumor treatment (Figure 8E).

### 3.5. The Inhibition of Metastasis

The lung commonly underwent metastasis for the 4T1 tumor-bearing mice. As demonstrated in Figure 8F, the reduction in breast cancer metastatic nodules in the lungs was observed in the DOX/PLGA, DOX/CAT@PLGA, and DOX/CAT@PLGA-M1 groups compared with the mice in the control group (*p* < 0.01). The DOX/CAT@PLGA-M1 group showed the fewest lung metastatic nodules in breast cancer, exhibiting the best anti-tumor metastasis effect. This was explained by the fact that CAT enhanced TME hypoxia and reduced distant tumor metastasis. On the other hand, the M1 macrophage membrane deepened the therapeutic effect on tumors and further reduced the ability of tumor metastasis [27]. The H and E staining of lung tissue displayed clear tumor metastatic nodules in the control group, and nodules persisted in the DOX group and DOX/PLGA group, indicating that the lung tissue was damaged. In the DOX/CAT@PLGA and DOX/CAT@PLGA-M1 groups, the number of pulmonary nodules significantly decreased, further verifying that the DOX/CAT@PLGA-M1 nanoparticles possessed the ability to significantly inhibit lung metastasis (Figure 9).

Biosafety is an important consideration in anticancer therapy. To further evaluate the in vivo biosafety of our method, the histology and morphology of major organs were compared after the treatment of DOX/CAT@PLGA-M1 (Figure 9). Heart slices showed significant cardiotoxicity in the DOX group, which indicated that the NPs could reduce the cardiotoxicity of DOX. Other tissues and organs possessed no obvious toxicity, which further proves the safety of the NPs. The results indicated that DOX/CAT@PLGA-M1 had good biocompatibility and safety.

## 4. Conclusions

In conclusion, we successfully developed a novel class of M1 macrophage membrane-camouflaged nanoparticles based on CAT to relieve tumor hypoxia, which exhibited significant metastasis inhibition and an excellent antitumor effect. DOX/CAT@PLGA-M1 utilized tumor vasculature and internalized through α4–VCAM-1 interaction, and also efficiently accumulated at tumor sites. In the tumor-hypoxic microenvironment, intra-nanoparticle CAT could be activated by the high concentration of H_2_O_2_ to generate O_2_ in situ to alleviate tumor hypoxia, improving the TME, and resulting in boosting anti-tumor chemotherapy and inhibiting tumor metastasis. Here, we provided a potential combination therapy for antitumor treatment.

## Figures and Tables

**Figure 1 pharmaceutics-15-02243-f001:**
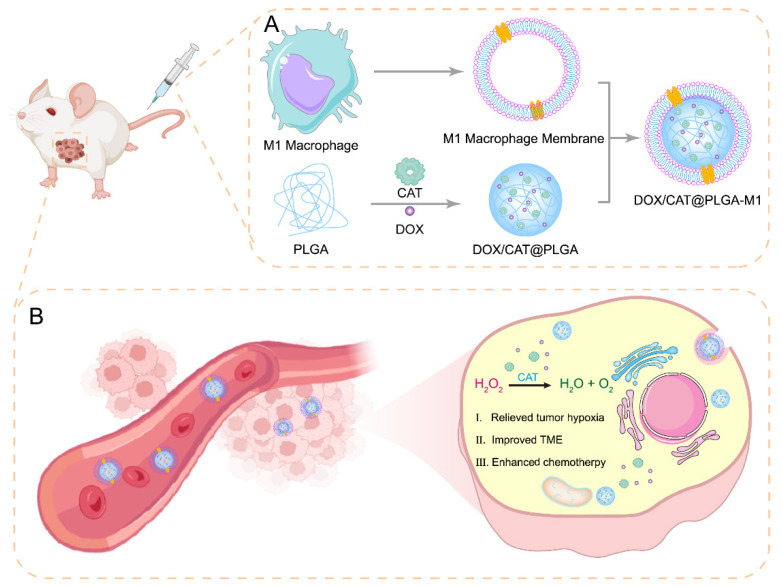
(**A**) Schematic illustration of the preparation of DOX/CAT@PLGA-M1 and (**B**) mechanisms of DOX/CAT@PLGA-M1 accumulation at the tumor site and release of O_2_ in situ by CAT to improve tumor microenvironment.

**Figure 2 pharmaceutics-15-02243-f002:**
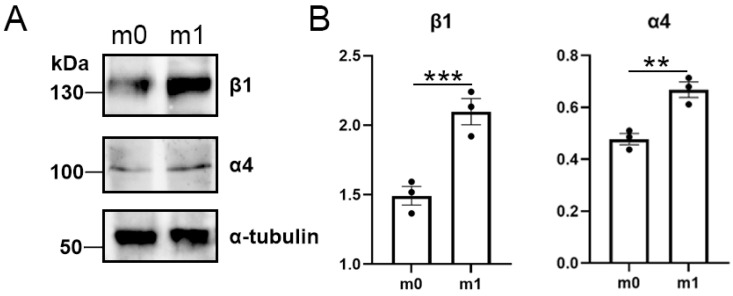
(**A**) Western blot analysis of β1 and α4 expression in PBNs after different treatments. (**B**) Expression levels of β1 and α4 after different treatments (*n* = 3) (** *p* < 0.01, and *** *p* < 0.001).

**Figure 3 pharmaceutics-15-02243-f003:**
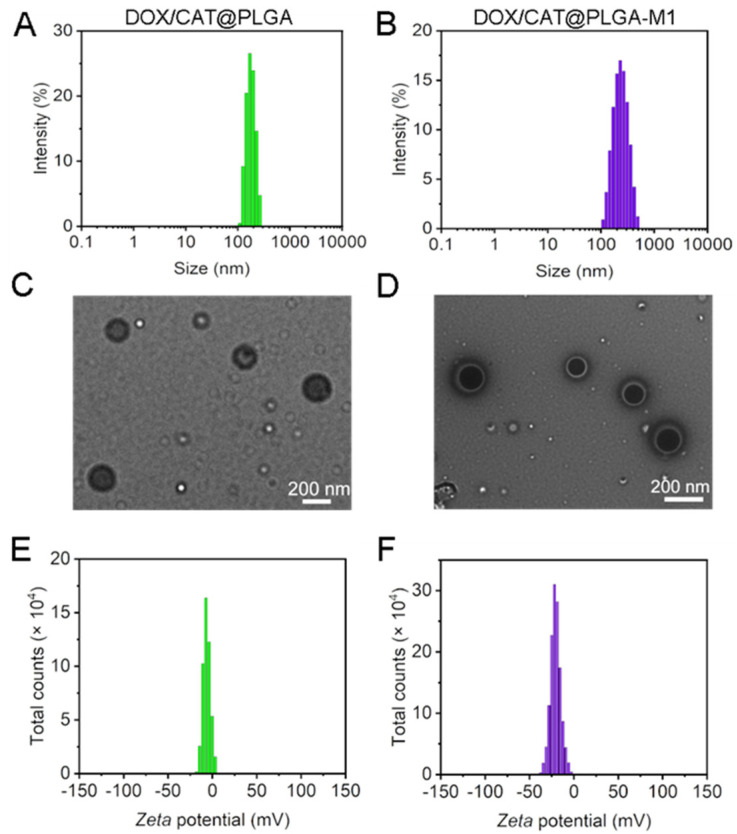
Particle size distribution, TEM images, and zeta potential of DOX/CAT@PLGA (**A**,**C**,**E**) and DOX/CAT@PLGA−M1 (**B**,**D**,**F**).

**Figure 4 pharmaceutics-15-02243-f004:**
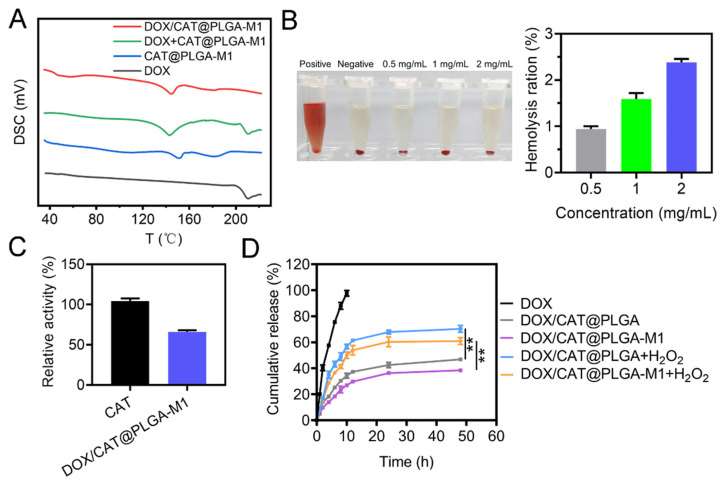
(**A**) DSC curves of free DOX, CAT@PLGA-M1, DOX + CAT@PLGA-M1, and DOX/CAT@PLGA-M1. (**B**) Hemolysis rate of DOX/CAT@PLGA-M1. (**C**) The CAT activities of DOX/CAT@PLGA-M1. (**D**) Release characteristics of DOX from free DOX, DOX/CAT@PLGA, and DOX/CAT@PLGA-M1 (*n* = 3) (** *p* < 0.01).

**Figure 5 pharmaceutics-15-02243-f005:**
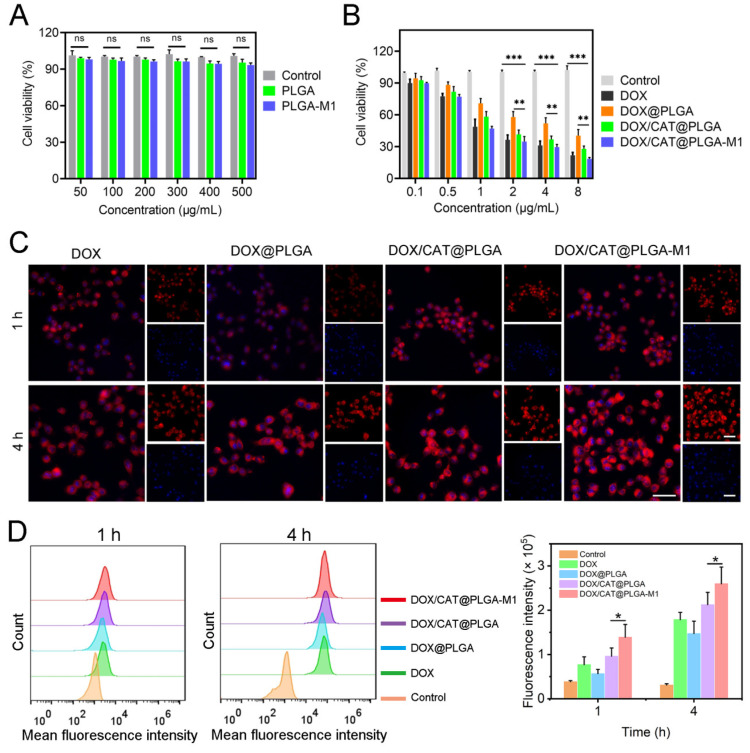
(**A**) Cell viability of blank carriers at different concentrations. (**B**) Cytotoxicity of different NPs. (**C**,**D**) Cellular uptake of different NPs at 1 h and 4 h by CLSM and flow cytometry (Scale bar: 50 μm) (* *p* < 0.05, ** *p* < 0.01, and *** *p* < 0.001).

**Figure 6 pharmaceutics-15-02243-f006:**
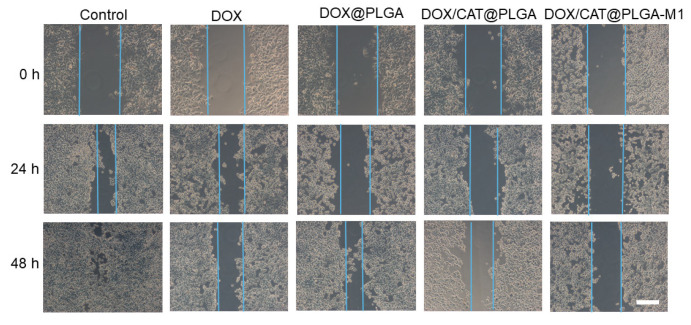
The migration images of 4T1 cells after different treatments at 0, 24, and 48 h (Scale bar: 200 μm).

**Figure 7 pharmaceutics-15-02243-f007:**
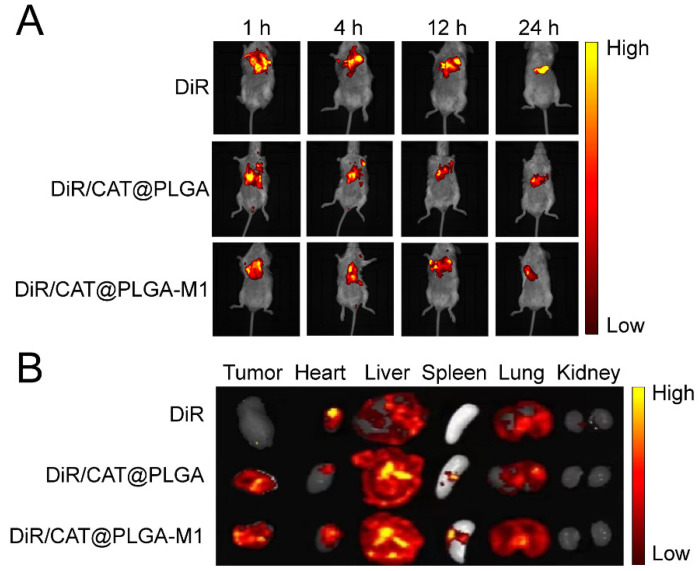
(**A**) Fluorescence imaging of NP distribution in 4T1 tumor-bearing mice in vivo. (**B**) Fluorescence images of major organs and tumor tissues at 24 h.

**Figure 8 pharmaceutics-15-02243-f008:**
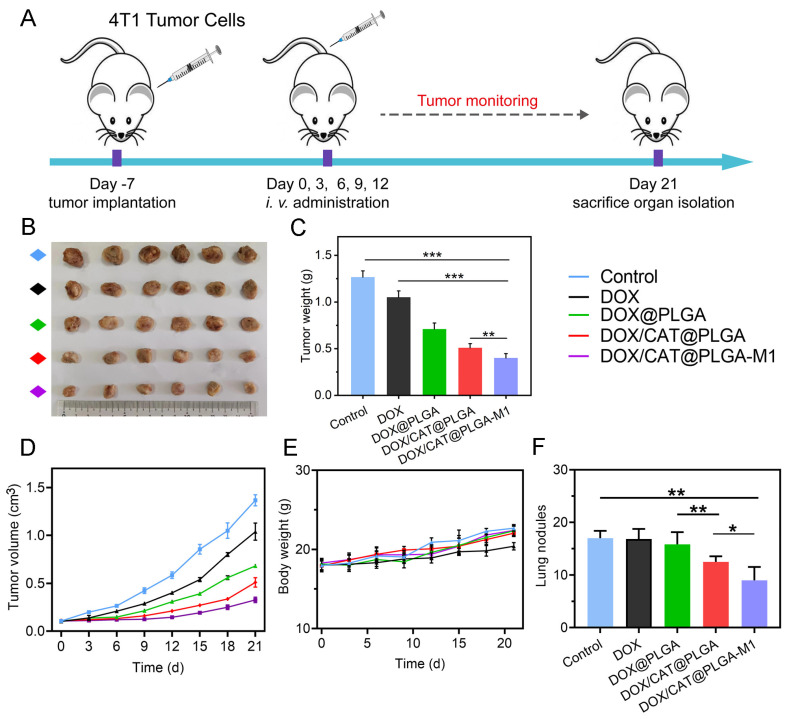
In vivo antitumor efficacy of 4T1 tumor-bearing mice. (**A**) The timeline of the efficacy study. (**B**,**C**) Typical tumor photographs and weight on day 21 post-treatment. (**D**) Relative tumor volume curves. (**E**) Body weight for all groups. (**F**) Lung nodule count of different groups (* *p* < 0.05, ** *p* < 0.01, and *** *p* < 0.001).

**Figure 9 pharmaceutics-15-02243-f009:**
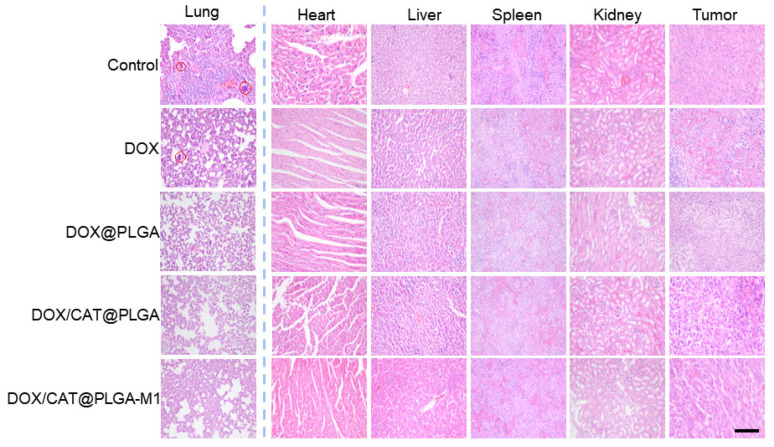
H and E staining of major organs and tumor tissues for different groups (Scale bar: 200 μm).

## Data Availability

Not applicable.

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
