# Peer review of "M1 Macrophage-Biomimetic Targeted Nanoparticles Containing Oxygen Self-Supplied Enzyme for Enhancing the Chemotherapy"

_pharmaceutics, 2023, doi:10.3390/pharmaceutics15092243_

Round 1
Reviewer 1 Report
1. I recommend the authors revise the abstract to highlight the key features of the nanoplate more prominently. Currently, about half of the abstract is dedicated to introducing the background or rationale of the study, while the essential aspects of the nanoplate are not fully presented.
2. Line 116-line 120, please include the catalase information in the description of DOX/CAT@PLGA preparation.
3. In Section 3.5, the authors mentioned a significant decrease in pulmonary nodules in both the DOX/CAT@PLGA and DOX/CAT@PLGA-M1 groups. However, to further strengthen the study, I recommend adding quantitative data, i.e., the average number of pulmonary nodules (in each image) of five groups. Additionally, it would be helpful to mark the pulmonary nodules in Figure 9 for better visualization and clarity.
4. Abbreviation. Please ensure that abbreviations, such as CAT, are defined only once when they first appear in the main body of the manuscript.
The study was well-designed and well-executed. Overall, the writing was easy to follow, although some parts were a little lengthy and wordy. The authors have provided ample information, making it possible for other researchers to repeat the experiment successfully.
Author Response
The reply was uploaded.

Reviewer 2 Report
The team demonstrates that PLGA nanoparticles contain M1 macrophage membrane, catalase and doxorubicin exerted better anti-tumor effects compared with PLGA-doxoribucin or free form of doxorubicin. Below are my comments:
My main concern is the hypoxic status in both in vitro and in vivo setting. The team claimed that catalase could alleviate the hypoxic condition by converting hydrogen peroxidase to oxygen. However, they never evaluate the hypoxic condition. I would like to suggest them compare the HIF-α expression levels between Control, DOX, DOX@PLGA, DOX/Cat@PLGA, and DOX/CAT/PLGA-M1 by western blotting (lysates from cell lines and tumor tissue) and IHC staining of tissue sample. The results could indicate the role of catalase in alleviating hypoxia.
Minor comments:
a) Each figure legend should include a short title to summarize the whole content.
b) Figure 2: compare 2 independent groups, the test should be student -T test or Mann-Whitney U test. One-way ANOVA is used to compare means for three or more groups.
c) Figure 4: comparative analysis like in (B) and (C), statistical analysis should be performed.
d) Figure 5: comparative analysis like in (B) and (D), statistical analysis should be performed.
e) Figure 7: please change “tumer” to tumor.
f) Figure 8C: please add a bar graph that compares the tumor weight with statistical analysis.
g) Figure 9: it is confusing to mix metastasis and drug toxicity together. Please separate them into 2 panels.
Minor English checking is required.
Author Response
The reply was uploaded.

Reviewer 3 Report
1. Has the stability of DOX under the influence of ultrasound been tested? 2. Section 2.2.1. is unclear. Where does catalase suddenly appear from? 3. RC-6 dissolution instrument - I do not know the use of the RC-6 dissolution instrument to study the release of substances from liposomes. Based on a quick analysis of data, it does not seem to me to be the right device for this purpose. Can you provide a paper that validated this method in a similar study? 4. The particle size is too large for parenteral use. How do the authors address this issue? 5. PDI is on the verge of being acceptable for medical purposes (see https://www.ncbi.nlm.nih.gov/pmc/articles/PMC6027495/) Release tests indicate that the drug is released almost completely in the first 12 hours. In the authors' opinion, is the formulation suitable for ex tempore execution by medical personnel?
Author Response
The reply was uploaded.

Round 2
Reviewer 3 Report
Paper can be published